# Parthenolide Has Negative Effects on In Vitro Enhanced Osteogenic Phenotypes by Inflammatory Cytokine TNF-α via Inhibiting JNK Signaling

**DOI:** 10.3390/ijms21155433

**Published:** 2020-07-30

**Authors:** Jin-Ho Park, Young-Hoon Kang, Sun-Chul Hwang, Se Heang Oh, June-Ho Byun

**Affiliations:** 1Department of Oral and Maxillofacial Surgery, Gyeongsang National University School of Medicine and Gyeongsang National University Hospital, Institute of Health Sciences, Gyeongsang National University, Jinju 52727, Korea; jinho.park@gnu.ac.kr (J.-H.P.); omfs00@gnu.ac.kr (Y.-H.K.); 2Department of Oral and Maxillofacial Surgery, Gyeongsang National University School of Medicine and Changwon Gyeongsang National University Hospital, Institute of Health Sciences, Gyeongsang National University, Jinju 52727, Korea; 3Department of Orthopaedic Surgery, Institute of Health Sciences, School of Medicine, Gyeongsang National University, Jinju 52727, Korea; hscspine@gnu.ac.kr; 4Department of Pharmaceutical Engineering, Dankook University, Cheonan 31116, Korea

**Keywords:** parthenolide, TNF-α, periosteum-derived cells, osteoblastic differentiation, JNK signaling

## Abstract

Nuclear factor kappa B (NF-κB) regulates inflammatory gene expression and represents a likely target for novel disease treatment approaches, including skeletal disorders. Several plant-derived sesquiterpene lactones can inhibit the activation of NF-κB. Parthenolide (PTL) is an abundant sesquiterpene lactone, found in Mexican Indian Asteraceae family plants, with reported anti-inflammatory activity, through the inhibition of a common step in the NF-κB activation pathway. This study examined the effects of PTL on the enhanced, in vitro, osteogenic phenotypes of human periosteum-derived cells (hPDCs), mediated by the inflammatory cytokine tumor necrosis factor (TNF)-α. PTL had no significant effects on hPDC viability or osteoblastic activities, whereas TNF-α had positive effects on the in vitro osteoblastic differentiation of hPDCs. c-Jun N-terminal kinase (JNK) signaling played an important role in the enhanced osteoblastic differentiation of TNF-α-treated hPDCs. Treatment with 1 µM PTL did not affect TNF-α-treated hPDCs; however, 5 and 10 µM PTL treatment decreased the histochemical detection and activity of alkaline phosphatase (ALP), alizarin red-positive mineralization, and the expression of ALP and osteocalcin mRNA. JNK phosphorylation decreased significantly in TNF-α-treated hPDCs pretreated with PTL. These results suggested that PTL exerts negative effects on the increased osteoblastic differentiation of TNF-α-treated hPDCs by inhibiting JNK signaling.

## 1. Introduction

Stromal stem cells possess the characteristic of “stemness”, defined as the capacity for self-renewal and the ability to generate differentiated progeny and produce diverse cell types. However, stemness has been shown to decrease with age and with increasing passage numbers, associated with molecular and biological age-associated changes, such as telomere dysfunction, decreased resistance to oxidative stress, disrupted mitochondrial metabolism, and inflammation [1,2,3,4]. Recently, the supplementation and additional management of mesenchymal stem cells (MSCs), including the pretreatment of MSCs with growth factors, such as bone morphogenetic proteins (BMPs) or pro-regenerative cytokines, such as transforming growth factor β (TGF-β), have been demonstrated to enhance the abilities of MSCs to differentiate into osteoblasts, which is necessary to improve the likelihood of using MSCs for tissue repair [5,6,7,8].

During bone regeneration, inflammation has great potential to regulate MSC osteogenesis. Although uncontrolled inflammation often has destructive effects on bone, several examples of inflammatory processes have been shown to trigger new bone formation. Tumor necrosis factor (TNF)-α is a major inflammatory cytokine that has been associated with bone loss in many inflammatory diseases. The influence of TNF-α on the osteoblastic differentiation of osteoprecursor cells has been reported to actively direct MSCs away from an osteoblastic fate [9,10,11]. However, the stimulatory effects of this cytokine, including the recruitment of MSCs and osteoblasts, have also been reported [12,13,14].

Parthenolide (PTL), isolated from the Mexican Indian Asteraceae family plants, induces apoptosis in several cell types, by inhibiting the nuclear factor-kappa B (NF-κB) signaling pathway [15,16]. Because the downstream signaling events of inflammatory cytokines include the activation of NF-κB, PTL may potentially serve as a novel treatment for inflammatory diseases. NF-κB signaling leads to the induction of osteoclast differentiation genes, the prolonged survival of osteoclasts, and the repression of osteoblastic differentiation in osteoblastic cell lines; therefore, PTL may have favorable activities toward the osteoblastic differentiation of MSCs. However, although PTL may play a role in the prevention of osteoclast formation, under inflammatory conditions, whether PTL can be used for bone formation and the osteoblastic differentiation of MSCs remains unclear.

Although osteogenic cells that are capable of forming bones, in vitro, can be obtained from several sources, the periosteum represents a good source for the preparation of appropriate osteogenic stem/progenitor cells that can be used therapeutically. Our previous work demonstrated that these human periosteum-derived cells (hPDCs) were positive for markers present in MSCs (CD44, CD73, CD90, and CD105) and expressed the pluripotent transcriptional factors Oct4, Nanog, and Sox-2. Moreover, these hPDCs were successfully differentiated into mesenchymal lineages, particularly adipocytes, osteocytes, and chondrocytes; thus, they can be considered to be human periosteum-derived MSCs [17,18].

Several mitogen-activated protein kinases (MAPKs), such as extracellular signaling-related kinase (ERK), c-Jun N-terminal kinase (JNK), and p38 MAPK, have been shown to favor osteoblastic cell differentiation and to activate osteoblast-specific gene expression, in several osteoprecursor cells. MAPK pathways are activated by environmental stresses, such as ultraviolet irradiation, heat and osmotic shock, genotoxic agents, anisomycin, and toxins, but can also be affected by growth factors and several cytokines, including PTL. PTL inhibits NF-κB, which has been shown to block osteoclastogenesis; however, PTL can also inhibit MAPKs and other kinases; therefore, the role played by PTL on individual MAPK signaling pathways during the osteoblastic differentiation of stem/progenitor cells remains unclear [19,20,21,22].

To our knowledge, limited evidence exists regarding the effects of PTL on the osteoblastic differentiation of hPDCs or TNF-α-treated hPDCs. In addition, the effects of PTL on the osteoblastic differentiation of TNF-α-treated hPDCs, which are mediated by the MAPK pathways, may also be controversial. This study aimed to examine whether PTL has specific effects on the osteogenic phenotypes of hPDCs or TNF-α-treated hPDCs and to examine whether any of three MAPK signaling pathways are associated with the effects of PTL on the osteoblastogenesis of TNF-α-treated hPDCs.

## 2. Results

### 2.1. Treatment of hPDCs with PTL

Because PTL is well-known to inhibit NF-κB activation and, thus, induce apoptotic cell death in different cell types, we examined the changes in hPDC viability following exposure to PTL, at various doses and times. PTL, at 10 µM, appeared to increase the viability of hPDCs after 3 days of culture and tended to decrease cell viability after 7 days; however, we observed no significant differences in the viability of hPDCs associated with different concentrations of PTL, regardless of the culture period (Figure 1A). In addition, because the NF-κB signaling pathway has been reported to be a negative regulator of osteogenesis and the suppression of the NF-κB signaling pathway can potently augment osteoblast differentiation, we also investigated the effects of PTL on the in vitro osteoblastic activity of hPDCs. Although PTL at 10 µM appeared to decrease the histochemical detection of alkaline phosphatase (ALP) and the alizarin red-positive mineralization in cells, treatment with PTL did not have significant effects on the bioactivity of ALP, the level of alizarin red-positive mineralization, or the calcium contents in hPDCs, regardless of the PTL concentration used (Figure 1B). These results suggested that PTL has no direct effects on the viability and osteoblastic activities of hPDCs under in vitro cell culture conditions, without the activation of NF-κB signaling.

### 2.2. Treatment of hPDCs with TNF-α

Inflammatory cytokine-mediated signals can activate several parallel and interconnected signaling cascades, resulting in the activation of NF-κB and PTL inhibits a common step that is associated with multiple pathways. Therefore, we first examined the effects of the inflammatory cytokine TNF-α on the in vitro osteoblastic differentiation of hPDCs, to better understand the biological role played by PTL. The histochemical detection and bioactivity of ALP were significantly increased in hPDCs treated with 10 ng/mL TNF-α (Figure 2A). Similar to the effects of TNF-α on ALP in hPDCs, alizarin red-positive mineralization and calcium contents were also markedly increased in TNF-α-treated-hPDCs (Figure 2B,C). These results suggested that TNF-α has positive effects on the in vitro osteoblastic differentiation of hPDCs.

### 2.3. Effects of PTL on the In Vitro Osteoblastic Phenotypes of hPDCs Treated with TNF-α

We next examined the effects of PTL on the in vitro osteoblastic phenotypes of hPDCs treated with TNF-α. Although treatment with 1 µM PTL did not affect the histochemical detection and activity of ALP in TNF-α-treated hPDCs, treatment with 5 and 10 µM PTL significantly decreased ALP expression and activity in TNF-α-treated hPDCs. Caffeic acid phenethyl ester (CAPE), used as a positive control, also decreased the histochemical detection and activity of ALP in these cells (Figure 3A). Similarly, although treatment with 1 µM PTL did not decrease the quantification of alizarin red-positive mineralization in hPDCs treated with TNF-α, the alizarin red-positive mineralization was decreased in cells pretreated with 5 and 10 µM PTL, in a concentration-dependent manner. CAPE also decreased alizarin red-positive mineralization, in a dose-dependent manner (Figure 3B). CAPE appeared to reduce ALP activity and alizarin red-positive mineralization to a slightly greater extent than PTL in hPDCs treated with TNF-α.

Baseline ALP mRNA expression was largely elevated at 2 weeks and gradually decreased with increasing culture time. Treatment with 5 and 10 µM PTL significantly decreased ALP expression at 1 and 2 weeks of culture; however, PTL had no effects beyond TNF-α-induced ALP mRNA at 4 weeks, regardless of concentrations. Baseline osteocalcin mRNA was expressed very weakly at 1 week, after which, expression increased in a time-dependent manner, throughout the duration of the culture. At 4 weeks, treatment with 5 and 10 µM PTL markedly decreased osteocalcin expression levels, whereas treatment with PTL had no additional effects on osteocalcin expression beyond those induced by TNF-α at 1 week and 2 weeks of culture; however, treatment with 10 µM PTL decreased the TNF-α-induced osteocalcin mRNA expression at 2 weeks of culture. Baseline expression levels of Runx2 was increased over 2 weeks in culture and gradually decreased with culture time. Treatment with 5 and 10 µM PTL markedly decreased TNF-α-induced Runx2 expression during the entire duration of culture (Figure 3C). Our results suggested that PTL plays a detrimental role in the in vitro osteoblastic differentiation of hPDCs activated by the inflammatory cytokine TNF-α.

We also examined the role of PTL as the inhibitor of NF-κB signaling by Western blotting for the detection of phosphorylation of p65. p65 is one of the five components that form the NF-κB transcription factor family. The phosphorylation of p65 was highly expressed in hPDCs treated with TNF-α. In addition, PTL dramatically blocked the p65 phosphorylation in hPDCs treated with TNF-α (Figure 3D). These results suggest TNF-α activated NF-κB signaling and PTL functions as an inhibitor of NF-κB.

### 2.4. Effects of MAPK Inhibitors on In Vitro Osteoblastic Phenotypes of hPDCs Treated with TNF-α

MAPKs regulate key transcriptional events that mediate osteoblast differentiation. However, evidence regarding the effects of PTL on the inflammatory cytokine-mediated osteoblastic differentiation of MSCs has been limited. To analyze the roles played by individual MAPK signaling pathways in the mechanisms underlying the PTL-mediated reduction in osteogenic potential in hPDCs treated with TNF-α, we first examined whether MAPKs were activated in hPDCs treated with TNF-α. As shown in Figure 4A, ALP activity significantly decreased in hPDCs pretreated with the selective inhibitors of ERK, p38 MAPK, and JNK, respectively. Among the tested MAPK inhibitors, SP 600125, which is a JNK-specific inhibitor, resulted in the most obvious reductions in the histochemical detection and activity of ALP in the cells. As shown in Figure 4B, alizarin red-positive mineralization was only markedly decreased in the JNK-inhibitor-treated-hPDCs. These results indicated that the activation of MAPK signaling is involved in the TNF-α-mediated increase in ALP activity and alizarin red-positive mineralization in hPDCs. In addition, JNK signaling plays a more critical role in the enhanced osteoblastic differentiation of TNF-α-treated hPDCs compared with other MAPK pathways.

### 2.5. Phosphorylation of MAPKs by PTL

We next examined the three individual MAPK signaling pathways associated with the effects of PTL on the reduced osteoblastic phenotypes of hPDCs treated with TNF-α. As shown in Figure 5A, the treatment of hPDCs with TNF-α stimulated the phosphorylation of ERK and JNK, whereas p38 MAPK phosphorylation was almost not detectable. JNK phosphorylation increased in cells treated with TNF-α. However, in hPDCs pretreated with PTL, p38 and ERK phosphorylation were strongly observed. In contrast, JNK phosphorylation was transiently observed at 0.5 h, followed by almost no detection in cells pretreated with PTL. The relative phosphorylation ratio measured by densitometric quantification of Western blot band intensities showed that JNK phosphorylation was significantly decreased in TNF-α-treated hPDCs pretreated with PTL, whereas p38 MAPK phosphorylation increased in the TNF-α-treated hPDCs pretreated with PTL (Figure 5B). Combined with the effects of MAPK inhibitors on the in vitro osteoblastic differentiation of the hPDCs treated with TNF-α, our results suggested that the negative effects of PTL were primarily associated with the inhibition of JNK signaling.

## 3. Discussion

NF-κB is a major transcription factor that regulates the expression of inflammatory mediators and cytokines, and its activation plays a critical role in the initiation and development of inflammatory bone disease, suggesting that the inhibition of NF-κB signaling could represent a promising therapeutic target for the inhibition of osteolysis. Although the effects of NF-κB activation on osteoblast differentiation and bone formation under in vitro culture conditions remain under debate, in general, NF-κB activation has been implicated in the stimulation of osteoclast differentiation and the inhibition of osteogenic differentiation in MSCs [23,24,25,26]. Chang et al. [27] reported that the time- and stage-specific inactivation of NF-κB signaling in differentiated osteoblasts significantly increased trabecular bone mass and bone mineral density, without affecting osteoclast activities in young mice, and rescued bone loss in an ovariectomized adult mouse model. Moreover, inflammatory cytokine-induced-NF-κB activation was found to inhibit osteoblast differentiation by attenuating canonical β-catenin signaling [23].

Several plant-derived small molecules have been identified as potential inhibitors of the NF-κB pathway. PTL, a major sesquiterpene lactone derived from feverfew, has been reported to be a potent inhibitor of NF-κB, through the interference in a common step of the NF-κB activation pathway [15,16]. In the present study, we examined the effects of PTL on changes in cell viability and in vitro osteoblastic activity in hPDCs. However, PTL did not significantly affect the cell viability or osteoblastic activities of hPDCs under in vitro cell culture conditions, without NF-κB signaling activation. Although further study remains necessary to elucidate the role played by PTL on the osteoblastic differentiation of MSCs under in vitro cell culture conditions, the primary findings of this study suggested that the effects of PTL on the osteogenic phenotypes of hPDCs are dependent of NF-κB activation by some exogenous stimuli, such as inflammatory cytokines.

Through ligand-receptor interactions, inflammatory cytokines can evoke a series of interconnected events, ultimately leading to the activation of NF-κB. Because PTL exerts its effects by inhibiting inflammatory mediator-induced NF-κB activation, we first examined the effects of the inflammatory cytokine TNF-α on the in vitro osteoblastic differentiation of hPDCs. Interestingly, in the present study, TNF-α stimulated the in vitro osteoblastic differentiation of hPDCs by enhancing ALP activity and the mineralization process. Bone remodeling at sites affected by inflammatory processes is influenced by the expression of inflammatory cytokines, which are expressed by immune cells present within the inflamed tissues. Inflammatory cytokines, such as TNF-α and interleukin (IL)-1β, influence bone remodeling by favoring bone resorption, via osteoclast activation. TNF-α and IL-1β also contribute to decreased bone mineral density by inhibiting osteoblast differentiation and bone formation [28,29,30,31]. Although inflammatory cytokines play an important role in the promotion of bone resorption, via either the direct or indirect promotion of osteoclastogenesis, the effects of inflammatory cytokines on the osteoblastic differentiation of MSCs remain controversial, with regard to whether they are anabolic or catabolic factors for bone metabolism. Many studies have reported the reduced differentiation potential of MSCs treated with inflammatory cytokines, while there is increasing evidence indicating an important induction role played by inflammatory cytokine in osteogenic differentiation [9,10,11,12,13,14,32,33]. When examining the role played by TNF-α in the osteogenic differentiation of MSCs, Mountziaris et al. [34] demonstrated that a lower dose of TNF-α (0.1–5.0 ng/mL) inhibited mineralization, whereas a higher dose (50 ng/mL) enhanced mineralization, in dexamethasone-pretreated rat MSCs. Croes et al. [13] observed that early treatments with extracellular NF-κB stimulators, including TNF-α, lipopolysaccharide (LPS), and peptidoglycan, enhanced osteogenesis in human adipose tissue-derived or bone marrow-derived MSCs. Lu et al. [35] reported that human adipose tissue-derived MSCs, preconditioned with 1 ng/mL TNF-α for 1–3 days, showed enhanced osteogenic differentiation, via the induction of BMP-2 expression. Both positive and negative effects of cytokines on the osteogenic differentiation of MSCs have been reported, which might be related to the dose or the exposure times of the inflammatory cytokine, the species and physiological status of the donor cells, and the osteogenic induction conditions used [36].

We next examined the roles played by PTL in the determination of in vitro osteoblastic phenotypes in hPDCs treated with TNF-α. Although 1 µM PTL did not affect the in vitro osteoblastic phenotypes of hPDCs, high concentrations (5 and 10 µM) of PTL decreased the osteoblastic differentiation of hPDCs by decreasing the histochemical expression and activity of ALP, decreasing alizarin red-positive mineralization, and decreasing the expression levels of ALP (after 1 and 2 weeks) and osteocalcin mRNA (after 4 weeks). Although PTL had no direct effects on the osteogenic phenotypes of hPDCs under in vitro cell culture conditions, without exposure to inflammatory mediators, these results indicated that PTL has adverse effects on the in vitro osteoblastic differentiation of hPDCs activated by the inflammatory cytokine TNF-α. Although the roles played by inflammatory cytokines during osteoblastogenesis under in vitro cell conditions remain controversial, bone homeostasis and metabolism are heavily dependent on the inflammatory environment produced by inflammatory signals, and our results suggested that PTL could modify bone metabolism in an inflammatory environment and that the involvement of an NF-κB signaling inhibitor during the inflammation and healing processes could represent a promising strategy for the promotion of bone tissue regeneration in actual clinical environments.

Accumulating evidence indicates that MAPK activation plays a critical role in osteoblastic differentiation for several osteoprecursor cells and that extracellular stimuli, such as oxidative stress and inflammatory cytokines, may activate the MAPK pathway through serial phosphorylation. Interestingly, PTL shows dual stimulatory and inhibitory effects on MAPKs in various cell lines, and the effects of PTL on the osteoblastic differentiation of inflammatory cytokine-mediated MSCs remain disputed [19,20,21,22,37]. Therefore, in the present study, we examined whether MAPKs were involved in the reduction of osteogenic potential mediated by PTL in hPDCs treated with TNF-α. SP 600125, a specific inhibitor of JNK, showed the most negative effects on the osteoblastic differentiation of hPDCs treated with TNF-α. In addition, TNF-α treatment stimulated the phosphorylation of JNK in hPDCs. Based on the relative phosphorylation ratio of MAPKs, JNK phosphorylation was significantly decreased in TNF-α-treated hPDCs pretreated with PTL compared with cells treated without PTL.

Inflammatory cytokines and environmental stress have been reported to activate p38. In regard to the osteoblast differentiation of osteoprecursor cells, several studies have suggested that p38 MAPK has been found to be important in stimulating osteoblast differentiation in vitro by inducing the expression of osteoblast-specific genes and mineral deposition [38,39,40]. However, it remains unclear whether p38 MAPK signaling is involved in the osteogenesis mediated by inflammatory cytokines in osteoprecursor cells. In the present study, although SB2035801, a selective inhibitor of p38 MAPK, significantly decreased ALP activity in hPDCs treated with TNF-α, it did not affect the mineralization of TNF-α-treated hPDCs. In addition, the treatment of hPDCs with TNF-α did not stimulate the phosphorylation of p38 MAPK. However, interestingly, p38 MAPK phosphorylation increased in the TNF-α-treated hPDCs pretreated with PTL. Considering PTL has negative effects on the TNF-α-enhanced in vitro osteoblastic differentiation of hPDCs and decreases the phosphorylation of JNK stimulated by TNF-α treatment, despite the increase of p38 MAPK phosphorylation in the TNF-α-treated hPDCs pretreated with PTL, our results suggest that p38 MAPK does not seems to have some effects on the osteogenesis of TNF-α-treated hPDCs, The mechanism underlying osteoblastic differentiation and the activation of p38 phosphorylation by PTL in TNF-α-treated hPDCs requires further study.

Unfortunately, in the present study, the role of NF-κB in TNF-α-mediated osteogenic differentiation and the effects of PTL on NF-κB in TNF-α-mediated osteogenic differentiation have not been evaluated. However, considering the inflammatory cytokine TNF-α activated the osteogenic differentiation of hPDCs and also activated the NF-κB (p65) in hPDCs, we infer that NF-κB will be activated in the TNF-α-mediated osteogenic differentiation of hPDCs. Moreover, in the present study, PTL had no direct effects on the viability and osteoblastic activities of hPDCs under in vitro cell culture conditions, without the activation of NF-κB signaling. In view of those results, we think that PTL will inhibit activated NF-κB in the TNF-α-mediated osteogenic differentiation of hPDCs. In the further study, the role of NF-κB in TNF-α-mediated osteogenic differentiation and the potential effects of PTL on NF-κB activity could be identified to reveal the possibility of co-regulation by NF-κB and MAPK.

In conclusion, our results suggest that the functional effect of TNF-α in increasing the osteogenic phenotypes of hPDCs primarily depend on the JNK signaling pathway, and PTL treatment reveals the unfavorable effects on the enhanced osteoblastic differentiation of hPDCs treated with TNF-α, through the inhibition of JNK signaling.

## 4. Materials & Methods

### 4.1. Culture and Differentiation of hPDCs

Patients provided informed consent for the collection of periosteal tissues, as required by the Ethics Committee of Gyeongsang National University Hospital (GNUH 2014-05-012). Periosteal explants (5 × 20 mm) were harvested from mandibles during the surgical extraction of impacted lower third molars. hPDCs were isolated, as previously described [17,18]. Briefly, periosteal tissues were cultured in 100-mm culture dishes, in Dulbecco’s Modified Eagle Medium (DMEM), supplemented with 10% fetal bovine serum (FBS), 100 IU/mL penicillin, and 100 µg/mL streptomycin, at 37 °C, in 95% humidified air and 5% CO_2_. At 90% confluence, adherent cells were passaged by gentle trypsinization and reseeded in fresh medium. Osteoblast differentiation was induced by culturing periosteal cells (seeded at a density of 3 × 10^4^ cells/well in 24-well plates), at passage 3–5, in osteogenic induction medium, composed of DMEM, supplemented with 10% FBS, 50 µg/mL L-ascorbic acid 2-phosphate, 10 nM dexamethasone, and 10 mM β-glycerophosphate. The medium was changed every 3 days during the induction period.

### 4.2. Treatment of hPDCs with PTL

To investigate the effects of PTL on the viability and in vitro osteoblastic differentiation of hPDCs, hPDCs in osteogenic induction medium were treated with PTL (1, 5, and 10 µM, Sigma-Aldrich, St. Louis, MO, USA). The viability of the hPDCs was assayed using the Cell Counting Kit (CCK)-8 (Dojindo, Kumamato, Japan), using a previously published method [41]. The osteogenic phenotypes of hPDCs were also assessed, as previously described [17,18]. Runx2 and ALP are considered to be relatively early markers of osteoblast differentiation, whereas the secretion of osteocalcin, matrix mineralization, and calcium contents are associated with the late phase of osteoblast differentiation. Therefore, ALP staining and activity assays were performed at day 10 of culture, whereas the determination of alizarin red S staining and calcium contents were performed on day 21 of culture. Media were changed every 3 days, and PTL was added at each change of the medium.

### 4.3. Treatment of hPDCs with TNF-α

To evaluate the effects of the inflammatory cytokine TNF-α on the in vitro osteoblastic differentiation of hPDCs, periosteal cells that were cultured in osteogenic induction medium were treated with 10 ng/mL TNF-α (PeproTech, Rocky Hill, NJ, USA). The cytokine was also added at each change of the medium.

### 4.4. Effects of PTL on In Vitro Osteoblastic Phenotypes of hPDCs Treated with TNF-α

To examine the effects of PTL on the in vitro osteoblastic phenotypes of hPDCs treated with TNF-α, hPDCs were pretreated with PTL (1, 5, and 10 µM) in DMEM for 12 h before stimulation with TNF-α, and then the cells were treated with 10 ng/mL TNF-α. CAPE, another selective inhibitor of NF-κB signaling, was used as a positive control. The PTL and cytokine were also added at each change of the medium.

In addition, to evaluate the role of PTL as an inhibitor of NF-κB signaling, the expression of p65 was examined by Western blotting. To extract all proteins, hPDCs were placed in RIPA buffer (Cell Signaling Technology, Danvers, MA, USA), containing a protease and phosphatase inhibitor cocktail. After 20 min, the cell pellets were sonicated and centrifuged, and the supernatant was resolved by sodium dodecyl sulfate-polyacrylamide gel electrophoresis (SDS-PAGE), followed by the electrophoretic transferred onto a polyvinylidene difluoride membrane (Millipore, Burlington, MA, USA). The membranes were then probed with primary antibodies against phospho-p65 and β-actin (Cell Signaling Technology, MA, USA), respectively. Specific antibody binding was detected by horseradish peroxidase-conjugated secondary antibodies (Invitrogen, Carlsbad, CA, USA), and then visualized using an enhanced chemiluminescence detection reagent (Pierce Chemical Co, Rockford, IL, USA). Densitometry of the Western blot bands was performed using the image J software.

### 4.5. Real-Time Quantitative Polymerase Chain Reaction (qPCR) Analysis

The expression levels of osteoblast-related genes (ALP, osteocalcin, and Runx2) were analyzed in TNF-α-treated hPDCs, after pretreatment with PTL (1, 5, and 10 µM), at the indicated times. First-strand cDNA was generated using random hexamer primers, provided in the iScript cDNA Synthesis Kit (Bio-Rad Laboratories, Inc., Hercules, CA, USA). All primers and probes (ALPL #Hs010291444_m1; BGLAP #Hs01587814_g1; RUNX2 #Hs00231692_m1; GAPDH #Hs99999905-m1) were obtained commercially (TaqMan^®^ Gene Expression Assay, Applied Biosystems, Inc., Bedford, MA, USA) and amplified using a kit, following the manufacturer’s instructions (TaqMan Gene Expression Master Mix, Applied Biosystems). Amplification conditions were as follows: 50 °C for 2 min; 95 °C for 10 min; followed by 40 cycles of 94 °C for 15 s and 60 °C for 1 min, in 96-well plates using the ViiA 7 Real-Time PCR System (Applied Biosystems, Inc.). GAPDH was used as an internal control. All experiments were performed in triplicate.

### 4.6. Effects of MAPK Signaling Pathways on In Vitro Osteoblastic Differentiation of hPDCs Treated with TNF-α

To analyze the roles played by individual MAPK signaling pathways on the effects of PTL in hPDCs treated with TNF-α, we first examined whether MAPK signaling pathways were involved in the in vitro osteoblastic differentiation of hPDCs treated with TNF-α. The hPDCs were pretreated with 20 μM/mL PD98059 (Sigma-Aldrich, St. Louis, MO, USA), 10 μM/mL SB2035801 (Sigma-Aldrich, St. Louis, MO, USA), and 20 μM/mL SP600125 (Sigma-Aldrich, St. Louis, MO, USA), which are selective inhibitors of ERK, p38 MAPK, and JNK, respectively. Then, the cells were treated with 10 ng/mL TNF-α.

### 4.7. Phosphorylation of MAPKs by PTL

Next, we investigated the roles of three MAPK signaling pathways relative to the effects of PTL on hPDCs treated with TNF-α by Western blot analysis mentioned above. Primary antibodies against ERK, phospho-ERK, JNK, phospho-JNK, p38, phospho-p38, and anti-β-actin (all from Cell Signaling Technology, MA, USA) were used.

### 4.8. Statistical Analysis

Each experiment was performed independently, at least three times. The results of one of three independent experiments are shown, as representative data. Data are expressed as the mean standard deviation. Statistical analyses were computed using GraphPad Prism software (GraphPad Software, CA, USA). Data were evaluated using a one-way analysis of variance (ANOVA), with Tukey’s multiple comparisons and the Mann-Whitney U test. Comparisons with *p* < 0.05 were considered significant.

## Figures and Tables

**Figure 1 ijms-21-05433-f001:**
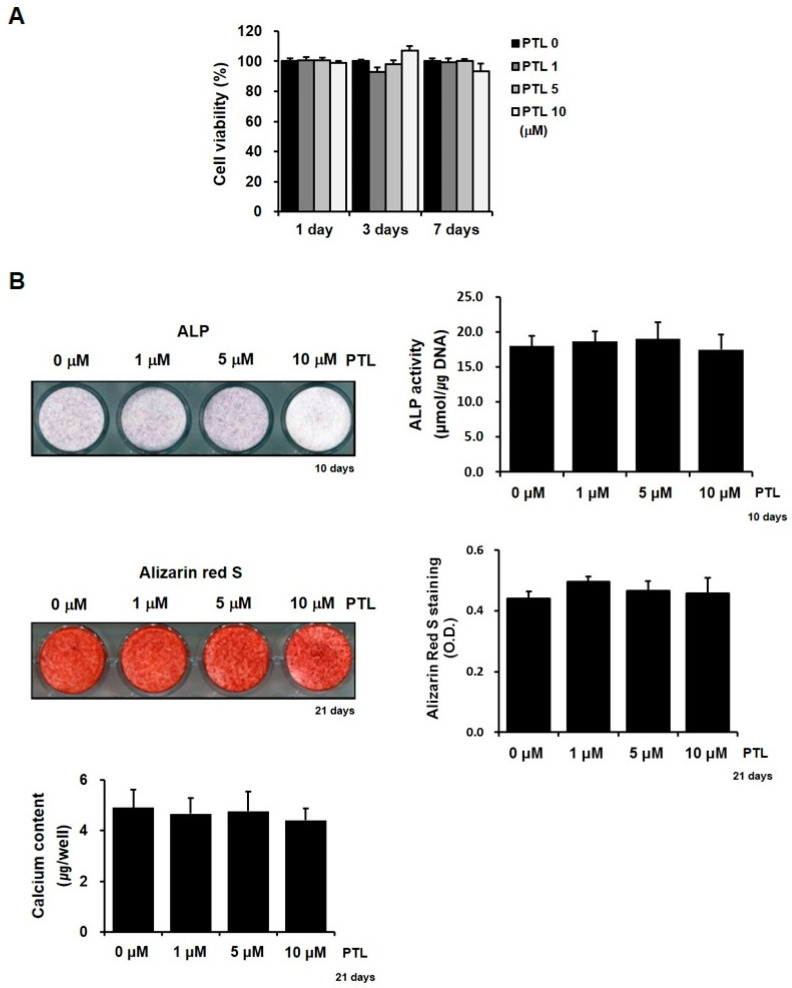
PTL has no significant effects on hPDCs in vitro. There was no significant effect of PTL on the cell viability of hPDCs (**A**). PTL also did not clearly affect the in vitro osteoblastic differentiation of hPDCs, regardless of concentrations (**B**).

**Figure 2 ijms-21-05433-f002:**
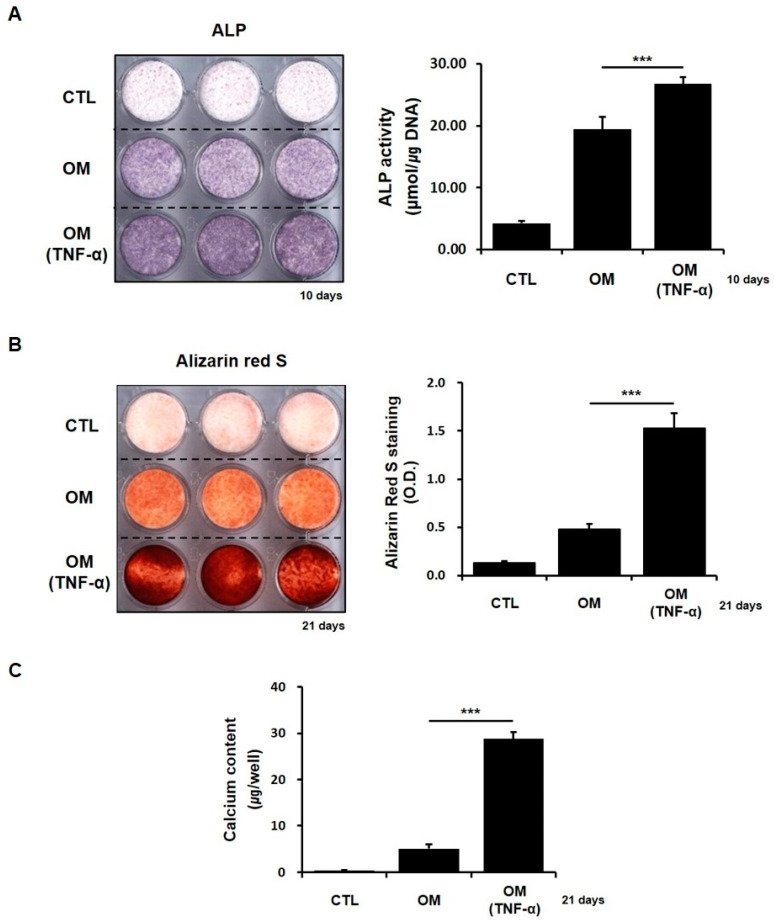
Treatment with 10 ng/mL TNF-α significantly increased the histochemical detection and activity of ALP in hPDCs (**A**). Alizarin red-positive mineralization and calcium contents also increased in hPDCs treated with 10 ng/mL TNF-α (**B**,**C**, respectively). CTL, control DMEM; OM, osteogenic induction medium. *** *p* < 0.001.

**Figure 3 ijms-21-05433-f003:**
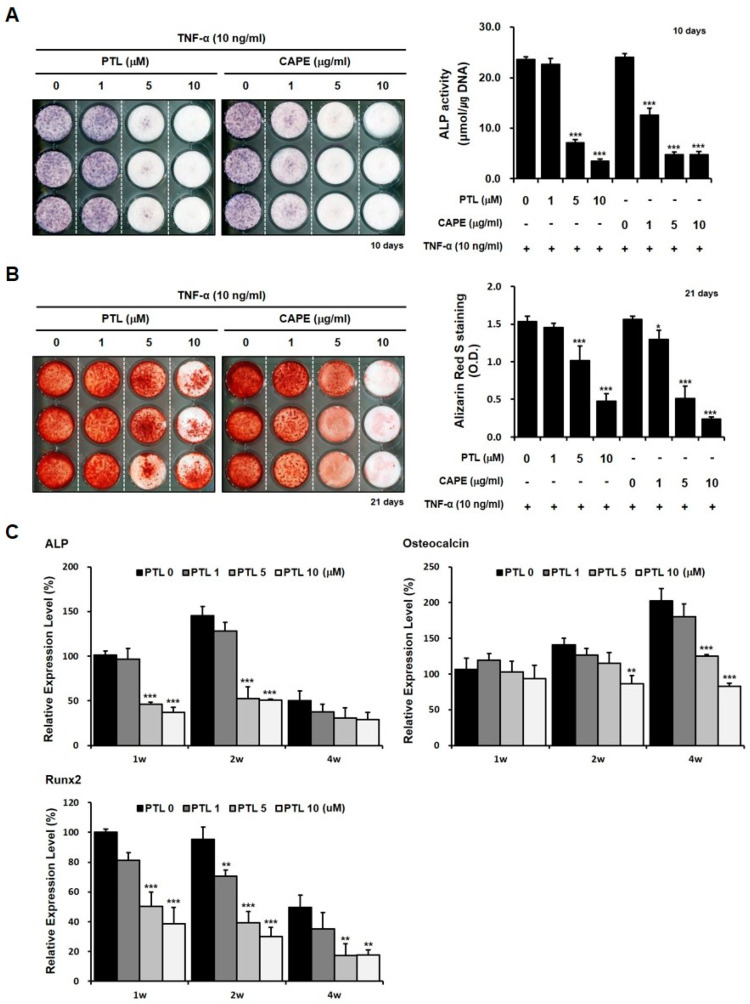
Effects of PTL on the osteogenic phenotypes of hPDCs treated with TNF-α. Treatment with 5 and 10 µM PTL significantly increased the histochemical detection and activity of ALP in hPDCs treated with TNF-α (**A**). The alizarin red-positive mineralization also appreciably decreased in hPDCs treated with 5 and 10 µM PTL (**B**). In addition, treatment with 5 and 10 µM PTL significantly decreased ALP mRNA expression in cells treated with TNF-α, at 1 and 2 weeks of culture, and decreased osteocalcin mRNA expression levels, at 4 weeks of culture, and decreased Runx2 mRNA expression levels, at 1, 2, and 4 weeks of culture (**C**). Treatment with TNF-α activated the phosphorylation of p65 in hPDCs, and PTL dramatically blocked the p65 phosphorylation in hPDCs treated with TNF-α (**D**). * *p* < 0.05. ** *p* < 0.01. *** *p* < 0.001.

**Figure 4 ijms-21-05433-f004:**
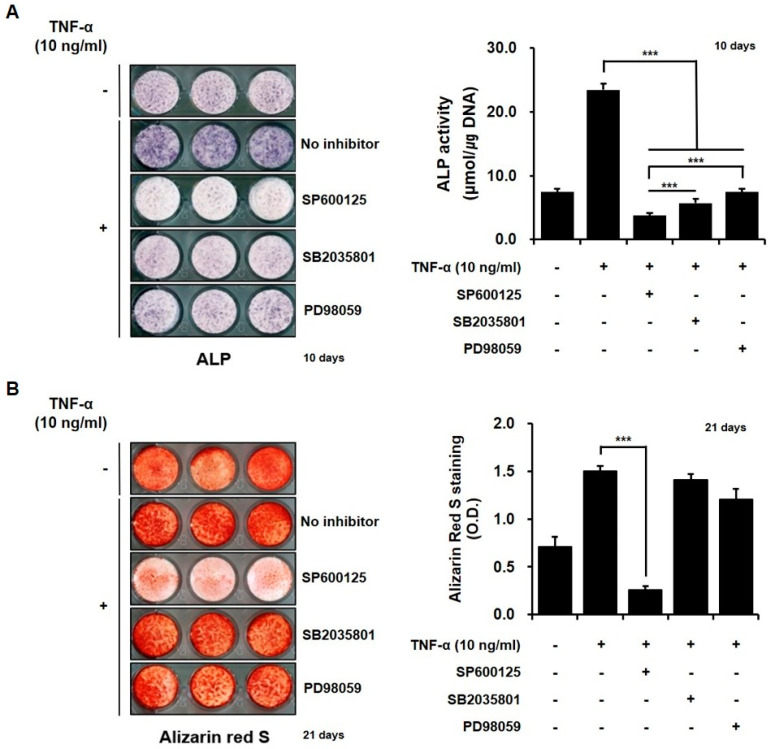
Effects of MAPKs on the in vitro osteoblastic differentiation of hPDCs treated with TNF-α. ALP activity decreases in hPDCs pretreated with all three MAPK inhibitors (**A**). SP 600125 shows the most potent inhibition of ALP activity, among all tested MAPK inhibitors. In line with the ALP results, alizarin red-positive mineralization was significantly decreased in hPDCs pretreated with the JNK-specific inhibitor (**B**). *** *p* < 0.001.

**Figure 5 ijms-21-05433-f005:**
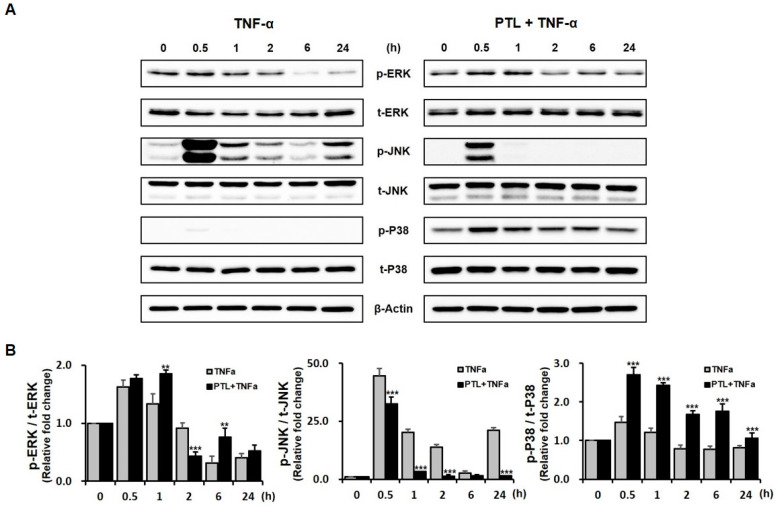
Effects of PTL on TNF-α-induced MAPK activation in hPDCs. TNF-α stimulated the phosphorylation of ERK and JNK in hPDCs, whereas pretreatment with PTL enhanced ERK and p38 phosphorylation in TNF-α-treated hPDCs. The effects of PTL on JNK phosphorylation were transient, at 0.5 h, in the TNF-α-treated hPDCs (**A**). The quantification of the Western blot was performed to determine the relative phosphorylation ratio of the MAPK pathway in hPDCs, which showed that the relative phosphorylation ratio of JNK was significantly decreased in TNF-α-treated hPDCs pretreated with PTL, whereas the relative phosphorylation ratio of p38 MAPK increased in cells pretreated with PTL. The phosphorylation of ERK significantly increased after 1 and 6 h, although it decreased at 2 h following pretreatment with PTL (**B**) ** *p* < 0.01, *** *p* < 0.001.

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
