# Peer review of "Parthenolide Has Negative Effects on In Vitro Enhanced Osteogenic Phenotypes by Inflammatory Cytokine TNF-α via Inhibiting JNK Signaling"

_ijms, 2020, doi:10.3390/ijms21155433_

Round 1

Reviewer 1 Report

Park and colleagues investigated the potential implication of Parthenolide (PTL) in the regulation of extracellular matrix (ECM) formation during cell differentiation. They specifically explored the effect of PTL on the human periosteum-derived cells (hPDCs) during the osteogenic differentiation that was enhanced by the inflammatory cytokine tumor necrosis factor (TNF)-a. They used classical techniques to monitor ECM formation. This study mainly showed that exposure of the TNF-a- treated hPDCs to PTL during the osteogenic differentiation leads to decreased ECM production due to the inhibition of the c-Jun N-terminal kinase (JNK) signaling.

The paper is well written and the experiments are well conducted.

In this manuscript, the duration of the osteogenic differentiation is not clearly stated. All paper relies on the differentiation procedure that must be identical for all experiments to allow accurate comparison and data interpretation. The author must state for each experiment they perform an osteogenic differentiation the duration used. Varying the duration may vary the ALP activity and/or the amount of ECM produced and detected by Alizarin.

The authors looked only for very few markers of the osteogenic differentiation. It is, however, intriguing not to investigate the variation of the master gene of the osteogenic differentiation Runx2, either using western blot and/or RTqPCR to further support the data.

Reviewer 2 Report

The study by Park and colleagues describes the effect of the plant-derived sesquiterpene parthenolide (PTL) on the osteogenic differentiation capacity of human periosteum-derived stem cells (hPDCs). The authors nicely show TNFα-dependent stimulation of osteogenic differentiation in hDPCs, which was significantly reduced after co-treatment with PTL. On molecular level, phosphorylation of JNK is demonstrated to be regulating osteogenic differentiation of hDPCs in a TNFα-dependent manner, while PTL efficiently blocks JNK-phosphorylation. Despite these interesting and promising data, the following major aspects have to be addressed by the authors prior to considering the present manuscript for publication:

Major comments

1.) Exposure of TNFα-treated hPDCs to PTL strongly increased the level of pP38 MAKP compared to hPDCs treated only with TNFα (Fig. 5A). How do the authors explain this phenomenon, which is not addressed sufficiently in the discussion section? This aspect is of particular importance, since with regards to these data JNK phosphorylation seems not the only major molecular mechanism regulating osteogenic differentiation in hPDCs as claimed by the authors. In addition, potential target genes of MAPK signaling influencing osteogenic differentiation should be assessed to prove the hypothesis of MAPK regulating differentiation in hPDCs.

2.) The authors state in the abstract, introduction and discussion section, that PTL is a potent blocker of NF-κB signaling. The authors further apply TNFα, the major stimulator of canonical NF-κB signaling, to increase the osteogenic differentiation capacity of hPDCs. However, the molecular mechanisms presented are focusing solely on MAPK signaling. The authors should provide novel data examining the role of NF-κB in TNF-α-mediated osteogenic differentiation and potential effects of PTL in this context. Here, both NF-κB activity and potential target genes specific for osteogenic differentiation should be assessed.

Minor comments

1.) Figure quality should be overall improved and distances between panels within a figure should be shortened.

2.) The discussion section focuses to a great extend on NF-κB signaling but only a small paragraph is discussing MAPK signaling in the context of the present data. The focus of the discussion section should be on MAPK signaling instead of NF-κB signaling in accordance to the presented findings.

3.) Lines 216-217: Font size should be revised.

Round 2

Reviewer 1 Report

No comment

Reviewer 2 Report

Major comment 1

Although Park and colleagues did not provide novel data on MAPK target gene expression, major comment 1 is sufficiently addressed.

Major comment 2

The new data provided nicely show an inhibition of NF-κB by PTL. However, the authors did not assess the role of NF-κB in TNF-α-mediated osteogenic differentiation and potential effects of PTL in this context, as initially requested. NF-κB activity and potential target gene expression specific for osteogenic differentiation should be assessed also during differentiation of hPDCs. This aspect is of particular importance, since PTL clearly affects both NF-κB activity and MAPK signaling. Thus, the observed effects of PTL on TNF-α-mediated osteogenic differentiation may be based on a co-regulation by the NF-κB and MAPK instead of MAPK alone.

Minor comments.

Minor comments are sufficiently addressed.

Round 3

Reviewer 2 Report

Although the data included during the first revision show an inhibition of NF-κB by PTL in undifferentiated hPDCs, the authors still did not assess the role of NF-κB in TNF-α-mediated osteogenic differentiation. Likewise, the authors did not assess potential effects of PTL on NF-κB-activity in the context of TNF-α-mediated osteogenic differentiation, as initially requested. With PTL being commonly described to inhibit NF-κB activity, investigating its potential involvement in osteogenic differentiation of hPDCs next to MAPK is absolutely necessary to verify the main conclusion of the manuscript. Here, the observed effects of PTL on TNF-α-mediated osteogenic differentiation may be based on a co-regulation by the NF-κB and MAPK instead of MAPK alone as currently claimed by the authors. This aspect is of particular importance, since the present data clearly show an inhibition of NF-κB activity and MAPK signaling by PTL. Despite my previous comments in the first revision, the authors did not clarify this highly relevant issue on experimental level at all.
